# Identification, Screening and Antibacterial Mechanism Analysis of Novel Antimicrobial Peptides from Sturgeon (*Acipenser ruthenus*) Spermary

**DOI:** 10.3390/md21070386

**Published:** 2023-06-29

**Authors:** Hai-Lan Li, Ya-Nan Chen, Jun Cai, Tao Liao, Xiao-Yan Zu

**Affiliations:** 1Key Laboratory of Cold Chain Logistics Technology for Agro-Product (Ministry of Agriculture and Rural Affairs), Institute of Agro-Products Processing and Nuclear Agricultural Technology, Hubei Academy of Agricultural Sciences, Wuhan 430064, China; hl.li@hbaas.com (H.-L.L.);; 2Key Laboratory of Fermentation Engineering (Ministry of Education), Hubei Provincial Cooperative Innovation Center of Industrial Fermentation, Hubei Key Laboratory of Industrial Microbiology, Hubei University of Technology, Wuhan 430068, China

**Keywords:** sturgeon spermary peptide, identification, molecular docking, antibacterial mechanism

## Abstract

Fish is an important source of antimicrobial peptides. This study aimed to identify and screen antibacterial peptides with excellent antibacterial activity derived from sturgeon spermary peptides (SSPs) and to analyze their antibacterial activity and mechanism. Liquid chromatography-mass spectrometry/mass spectrometry methods were used to analyze and identify peptide sequences, computational prediction tool and molecular docking methods were used for virtual screening of antimicrobial peptides, and finally, candidate peptides were synthesized by solid-phase synthesis method. The results demonstrate that SSPs have excellent inhibitory activity against *Escherichia coli* with an inhibitory rate of 76.46%. Most parts of the SSPs were derived from the sturgeon (*Acipenser ruthenus*) histones, and the coverage of histone H2B was the highest (45%). Two novel peptides (NDEELNKLM and RSSKRRQ) were obtained by in silico prediction tools and molecular docking, which may interact with the DNA gyrase and dihydrofolate reductase of *E. coli* by forming salt bridges and hydrogen bonds. Compared to the individual peptides, the antibacterial effect was significantly improved by mixing the two peptides in equal proportions. Two novel peptides change the permeability of the *E. coli* cell membranes and may exert antimicrobial activity by inhibiting the metabolic process of the nucleic acids.

## 1. Introduction

Bacterial pneumonia is a significant public health problem worldwide, and the control of its related pathogenic microorganisms is considered to be one of the major challenges in ensuring human safety [1,2]. *Staphylococcus aureus* and *Escherichia coli*, two representative Gram-positive and Gram-negative pathogens are common foodborne pathogens that can cause food poisoning and toxic shock syndrome [3]. *Pseudomonas aeruginosa*, a non-fermenting Gram-negative bacterium, is responsible for most hospital-acquired infections, along with *S. aureus* and *E. coli* [4]. In recent years, due to the adverse effects of chemical preservatives on human health and increasing resistance to antibiotics, a growing number of researchers have focused on natural antimicrobial agents. In this context, the extraction of low-molecular-weight antimicrobial peptides (AMPs) from various organisms has received significant attention [5,6].

AMPs, a class of structurally diverse peptides widely found in nature, exhibit broad-spectrum antibacterial activity both in vivo and in vitro. All multicellular organisms have an anti-pathogen defense mechanism that can produce a variety of AMPs through their innate immune system [7]. Fish are an important source of antimicrobial peptides because they express all major classes of AMPs including defensins, cathelicidins, hepcidins and histone-derived peptides [8]. Histones have a crucial role in the packaging and organization of DNA and chromatin within the nucleus of cells [9]. These are conserved, positively charged proteins. In several systems, it was also shown that histone-derived fragments carry antimicrobial activities in a diverse range of organisms (frog, shrimp, etc.) [9,10,11]. In recent years, histones and their derived peptide fragments have been shown to possess broad-spectrum antimicrobial properties with effective inhibitory activity against both human and fish pathogens [12,13,14]. A previous study reported that histone H2B-like proteins isolated from the skin of catfish exhibit broad-spectrum antibacterial activity and effectively inhibit the growth of *E. coli* D31, *Aeromonas hydrophila* and *Saprolegnia* [15]. The antimicrobial mechanism of histone-derived peptides is thought to be partly due to their membrane-penetrating activity, which enables them to kill microorganisms by entering the cell interior and interacting with nucleic acids, thus interfering with normal intracellular functions [10,16]. Intracellular action AMPs exert antibacterial effects by inhibiting antibacterial targets in cells [17] such as DNA gyrase and dihydrofolate reductase (DHFR). Previous studies have shown the inhibitory effect of antimicrobial peptide on both enzymes [18]. Studies have been conducted on histone-derived peptides with antibacterial activity found in the skin mucus of fish and toads [8,19], as well as histone-derived peptides in disk abalone [20,21], chicken [22] and scallop [23]. However, only a few studies on the antibacterial activity of histones derived from fish spermary tissue are found, including Atlantic salmon (*Salmo salar*) [24], olive flounder (*Paralichthys olivaceus*) [25] and sturgeon [26].

In this study, a sturgeon spermary protein extract (SSPE) was used as a raw material [26] to prepare sturgeon spermary peptides (SSPs) through enzymatic hydrolysis and purification. The amino acid sequences of SSPs were analyzed and identified by liquid chromatography-mass spectrometry/mass spectrometry (LC-MS/MS). Target peptides with good antibacterial activity potential were screened with the help of an antimicrobial activity prediction tool and molecular docking. Then, the target peptides were synthesized to verify their antibacterial activity and further elucidate their antibacterial mechanism. This study provides data support for the prediction, screening, and structure–activity relationship research of high-efficiency fish-derived antimicrobial peptides.

## 2. Results and Discussion

### 2.1. Antibacterial Activity of SSPs

Our previous study showed that SSPEs efficiently inhibit the growth of *E. coli* [26]. Papain was used to hydrolyze SSPEs to further explore its potential antibacterial fragments. The results demonstrate that the SSPs markedly inhibit the growth of *E. coli*, *S. aureus* and *P. aeruginosa* both before and after purification (Figure 1). The SSPs showed the highest inhibition rate and the largest inhibition zone (Figure 1B) against *E. coli*, with an inhibitory rate of 76.46% (Figure 1A). Antimicrobial peptides in fish have been extensively studied since a highly effective antimicrobial peptide was initially isolated from the secretion of Moses sole (*Pachygrapsus marmoratus*) [13,27]. Histones of fish are gradually replaced by protamine as the testicular tissue matures and only approximately 10% of histones are retained [28]. Similar to the results of this study, a protein extracted from the spermary of *P. olivaceus* is highly homologous to histone H1, and it exhibited antibacterial activity against *E. coli* D31 and *S. aureus*, with a minimum effective concentration of 1.6 μg/mL and 30 μg/mL, respectively [25].

### 2.2. Identification of SSPs

The peptide sequences of SSPs were identified by LC-MS/MS, and the results showed that SSPs were mainly derived from histones of sturgeon (*Acipenser ruthenus*) (Table 1). The SSPs contained fragments of different histones (H1, H2A, H2B and H4). Histone H2B showed the highest coverage (45%), followed by H2A (35%), but the latter had the higher score (Table 1). Papain is a specific proteolytic enzyme that hydrolyzes the carboxyl terminus of arginine and lysine in proteins and peptides, prefers an amino acid bearing a large hydrophobic side chain at the P2 position and does not accept valine in P1′ [29]. Previous studies successfully isolated six histone-derived peptides of the H2A family from the white blood cells of sturgeon (*Acipenser gueldenstaedtii*), among which the two N-terminal acetylated fragments showed good antibacterial activity against *E. coli* ML35p, *L. monocytogenes* EGD, methicillin-resistant *S. aureus* AATCC 33591 and *Candida albicans* [30].

To further explore the potential AMPs, sequence alignments of identified based on the Uniprot database were performed by BLAST of CAMPR4 database. The results show that three peptides in the histone H2B of *A. ruthenus*, KESYAIY, NSFVNDIFE and YNKRSTITS are completely matched with the fragments in the histone H2B of Schlegel’s green tree frog (*Rhacophorus schlegelii*) in the CAMPR4 database (Table 2), indicating their potential antibacterial activity. Previous studies have shown that the antibacterial activity of the histone H2B of *R. schlegelii* against *E. coli*, but not against *S. aureus*. [31]. Meanwhile, the eight peptides NDEELNK, NDEELNKL, NDEELNKLM, NDEELNKLMG, GVLPNIQ, GGVLPNIQ, GGVLPNIQA and IAQGGVLPNIQ from histone H2A are completely matched with the fragments in the histone H2B of Pacific white shrimp (*Litopenaeus vannamei*) in the CAMPR4 database (Table 2), indicating their potential antibacterial activity. Previous studies have shown that the histone and histone-derived peptide of *L. vannamei* exhibits excellent antibacterial activity against Gram-positive bacteria of *Micrococcus luteus* [32].

### 2.3. Virtual Screening of AMPs

Several antimicrobial peptide databases and prediction tools have been developed to facilitate the screening and evaluation of new antimicrobial peptides on a large scale [33]. In this study, two servers and three algorithms are used to predict potential antimicrobial peptides. Among the peptides obtained from de novo sequencing, the CPPpred score of 37 peptides is greater than 0.7, which are considered potential cell-penetrating peptides (Figure 2A). Subsequently, the results from CAMP_R4_ prediction showed that there are 11 peptides with the score of support vector machine and random forest greater than 0.5 and 0.45, respectively (Figure 2A). However, Among the peptides identified based on the Uniprot database, the CPPpred score of all peptides are less than 0.7, there are 25 peptides with a score of random forest greater than 0.45, while only 3 peptides remained after the BLAST with the CAMP_R4_ database (Figure 2B).

Previous studies provided effective evidence for AMPs serve as effective inhibitors of specific proteins of bacteria by molecular docking methods [17]. In the present study, 14 peptides were docked to two key receptor molecules of *E. coli* (DNA gyrase and DHFR). Based on the magnitude of the binding energy, the peptides with higher antibacterial potential were further screened. The results show that the binding energies of all 14 peptides to DNA gyrase were less than −5 kcal/mol (Figure 2C). However, the binding energies of KNQLLK, KTLKLK, KYLLK, NDEELNKLM and RSSKRRQ to the two enzymes were less than −5 kcal/mol (Figure 2C), indicating that they were more likely to bind to the two receptors while exhibiting stronger *E. coli* inhibitory activity. RSSKRRQ exhibited the lowest binding energy to both receptor molecules. Although the binding energy of NDEELNKLM with DNA gyrase was greater than that of other peptides, its binding energy to DHFR was the smallest (Figure 2C).

### 2.4. Molecular Docking Analysis of Peptides

The binding models of NDEELNKLM and RSSKRRQ peptides with DNA gyrase are shown in Figure 3A. NDEELNKLM contains a larger number of amino acids, and its GLU4, ASN6 and MET9 can form hydrogen bonds with the ASN269, GLN94 and ALA117 of DNA gyrase. In addition, the hydroxyl group of ASP2 forms a salt bridge and a hydrogen bond with the guanidinium group of the DNA gyrase residue ARG91. RSSKRRQ peptide contains more basic amino acids, and the guanidinium group of ARG1 forms a salt bridge with the ASP82(A) residue of the DNA gyrase. The carboxyl group of ARG1 can form hydrogen bonds with the hydrogen atoms of the guanidinium group in ARG121(C) and the hydroxyl group in TYR122(C) of the DNA gyrase residues. The guanidinium group of ARG6 forms a hydrogen bond with the carboxyl group of the DNA gyrase residue PRO79(A). The guanidinium group of ARG5 forms hydrogen bonds with the DNA gyrase residues ASP498(B), ASP500(B) and GLU424(B). Furthermore, the carboxyl groups of SER2 and SER3 can form hydrogen bonds with the amino acid residues of DNA gyrase.

The binding models of NDEELNKLM and RSSKRRQ peptides with DHFR are shown in Figure 3B. The hydroxyl group of GLU4 in NDEELNKLM peptide forms a hydrogen bond with the TRP23 amino hydrogen atom of the DHFR residue. The hydroxyl group of MET9 forms a hydrogen bond with the hydrogen atom of the hydroxyl group of the DHFR residue SER50. The guanidine group of ARG1 in RSSKRRQ peptide forms a salt bridge with the DHFR residue GLU28. The LYS4 in RSSKRRQ peptide forms hydrogen bonds with DHFR residues ILE21 and TRP23. Furthermore, the hydroxyl group of SER3 in RSSKRRQ peptide forms a hydrogen bond with the carboxyl group of SER50 in DHFR.

Both DNA gyrase and DHFR are important enzymes that are involved in regulating the process of intracellular nucleic acid metabolism [34,35]. Their inhibitors have long been considered effective candidate compounds for treating bacterial infections. The previous study has shown that inhibition of DHFR can lead to the death of pathogenic microorganisms, including *E. coli*, *M. tuberculosis* and *S. aureus,* by reducing the content of cellular tetrahydrofolate and interrupting DNA synthesis [36]. Meanwhile, the inhibition of DNA gyrase causes the breakage of microbial DNA single and double strands, disrupts the integrity of the genome and leads to cell death [35]. NDEELNKLM and RSSKRRQ tightly bind to DNA gyrase and DHFR by forming salt bridges and hydrogen bonds, suggesting that the inhibitory effects of these two peptides on *E*. *coli* may be mediated by the interaction between the peptides and the enzymes.

### 2.5. Antibacterial Activity Verification of Synthetic Peptides

The antibacterial activity of synthetic peptides against *E. coli* was verified. The results show that both NDEELNKLM and RSSKRRQ effectively inhibit the growth of *E. coli* (Figure 4A,B). However, NDEELNKLM has a more substantial inhibitory effect, with the IC_50_ of 2633.64 ± 700.91 μg/mL (Figure 4A). Moreover, a superior inhibitory effect on *E*. *coli* was obtained by mixing NDEELNKLM and RSSKRRQ in equal proportion, with the IC_50_ of 306.18 ± 68.81 μg/mL, which was approximately nine times lower than NDEELNKLM (Figure 4C). Histone-derived peptides from aquatic animals have been widely studied, Buforin I isolated from the stomach tissue of an Asian toad (*Bufo bufo gargarizans*), and Buforin II produced by the hydrolysis of Buforin I are both peptide fragments of the histone H2A and have been shown to possess broad-spectrum antibacterial activity [37,38]. Parasin I is a broad-spectrum antibacterial histone-derived peptide isolated from the skin of injured catfish that originates from an H2A precursor [19]. The presence of N-terminal basic amino acid residues in Parasin I was found to be crucial for its membrane-binding activity [39]. In the present study, NDEELNKLM derived from histone H2A showed antibacterial activity against *E. coli*. However, its amino acid sequence differs significantly from that of Buforin I and Parasin I derived from histone H2A. Unlike Buforin I and Parasin I, it is rich in acidic amino acids, suggesting that it may be a novel type of histone-derived peptide.

The antibacterial activity of fish spermary protein mainly originates from abundant arginine or lysine in its sequence [40]. However, few studies have been conducted on the antibacterial activity of hydrolysates from fish spermary protein. Iohara et al. [41] found that the VSRRRRRRRRGGRRRRR isolated from protamine had antifungal activity against *C. albicans*. Subsequently, Jun-ichi Nagao et al. [42] found that the antifungal activity of cyclic peptide (CRRRRRRGGRRRRC) against *C. albicans* was increased. The previous studies have shown that small-molecular-weight cationic peptides have the potential to become AMPs and that a net charge of +4 is sufficient to enable nine-amino acid peptides with antibacterial activity [43]. Moreover, small-molecular-weight cationic peptides may exhibit a unique mechanism of action that differs from that of large-molecular peptides [43]. In this study, RSSKRRQ is rich in arginine with a net charge of +4 (Figure 2A), which may act as a short cationic peptide to inhibit the growth of *E. coli*.

### 2.6. Antibacterial Mechanism Analysis of Synthetic Peptides

Because of its superior antimicrobial activity, the antibacterial mechanism of NDEELNKLM and RSSKRRQ mixture against *E. coli* was further explored, changes in membrane potential (Figure 5A) and effects on cellular viability (Figure 5B,C) were examined. Furthermore, the peptides localization analysis (Figure 5D) was conducted. bis-(1, 3-dibutylbarbituric acid) trimethine oxonol (DiBAC4(3)) and propidium iodide (PI) were used to assess changes in membrane potential and cell membrane integrity, respectively. Compared with the control group, the fluorescence intensity was significantly increased (*p* < 0.05) when the two peptides were mixed in equal proportions (Figure 5A,B). Moreover, with an increase in concentration, more red fluorescence was observed (Figure 5C). These results indicate that the combined action of the two peptides leads to a depolarization of the *E. coli* cell membrane, causing damage to the cell membrane and thereby affecting cell viability. The results of the laser scanning confocal microscope (LSCM) demonstrate that both NDEELNKLM and RSSKRRQ enter the interior of the *E. coli* cells and combine with the cell DNA (Figure 5D). Previous studies found that Buforin II exerts its antibacterial effects by crossing the cell membrane and interacting with intracellular targets such as DNA [44]. In contrast, Buforin I and Parasin I exert their antibacterial effect by disrupting the cell membrane [19,37]. However, these two antimicrobial mechanisms do not exist exclusively. Studies have shown that some AMPs may exert antibacterial effects through a dual antibacterial mechanism by simultaneously destroying the integrity of the cell membrane and inhibiting intracellular metabolism [45,46]. Interestingly, both NDEELNKLM and RSSKRRQ peptides may enter the interior of *E. coli* cells to exert antibacterial effects by affecting nucleic acid metabolism.

## 3. Materials and Methods

### 3.1. Materials

Sturgeon spermary tissue was purchased from Hubei Qingjiang Sturgeon Fishery Dragon Co., Ltd. (Yichang, China). Papain (9001-73-4, 6000 U/mg) was obtained from Sinopharm Chemical Reagent Co., Ltd. (Shanghai, China). *E. coli* CCTCC AB93154, *S. aureus* CCTCC AB2010020 and *P. aeruginosa* ATCC 27853 were obtained from the China Center for Type Culture Collection (CCTCC, Wuhan, China). Luria-Bertani Broth (LB) and Mueller Hinton Broth (MHB) medium were purchased from Hope Bio-Technology Co., Ltd. (Qingdao, China). Chromatographic grade reagents were used for LC and MS and purchased from Sigma-Aldrich (St. Louis, MO, USA). All other chemicals and reagents were analytical grade and purchased commercially.

### 3.2. Preparation of SSPs

The SSPE was prepared by referring to the method described previously [26]. Subsequently, the SSPE was dissolved in distilled water at a solid-to-liquid ratio (g/mL) of 1:50 and 0.65% (*w*/*v*) of papain was added. The pH was adjusted to 7, and the reaction was conducted at 37 °C for 6 h. The enzymatic hydrolysate was boiled for 10 min to stop the reaction and cooled to room temperature in an ice bath. The resulting enzymatic hydrolysate was centrifuged at 2000× *g* for 20 min at 4 °C, and part of the supernatant was collected and stored at −20 °C until further analysis. The remaining supernatant was filtered through a 0.22 μm filter membrane and the peptides were purified by a protein purification system (NGC Quest™ 10 Plus, Bio-Rad, Hercules, California, USA) equipped with Superdex Peptide 10/300 GL (*Φ* 1.0 cm × 30 cm, GE Healthcare, Boston, MA, USA). The peptides were eluted with a flow phase of 20 mmol/L PBS buffer (pH 7.0) at a flow rate of 0.5 mL/min and detected at a wavelength of 220 nm. All SSPs liquids containing the target peptides were collected. One portion was stored at −20 °C for further analysis, whereas the other was freeze dried using a freeze dryer (Gamma 1-16 LSC, Marin Christ, Germany) to obtain peptide powder.

### 3.3. Antibacterial Activity of SSPs

After obtaining two subcultures in LB medium, the strains were cultured with MHB medium at 37 °C and 60 r/min for 16–24 h until the viable bacterial count reached approximately 10^7^ CFU/mL. The supernatant of treated enzymatic hydrolysate was filtered with a 0.22 μm filter membrane, followed by the addition of 5.1 mL of sterile MHB, 0.3 mL of bacterial suspension and 0.6 mL of sample (using the same volume of MHB medium as control) to a 15 mL centrifuge tube. Then, they were incubated at 60 r/min and 37 °C for 12 h. Subsequently, 200 μL of the culture was transferred to a 96-well plate, and the absorbance (*A*) was measured at 600 nm using a microplate reader (Spark10M, Tecan, Mannedorf, Switzerland). The inhibition rate was calculated according to Equation (1).
(1)$$\mathrm{Inhibition}\;\mathrm{ratio}\left(\%\right)=\frac{A_{Control}-A_{SSPs}}{A_{Control}}\times 100\%.$$

In Equation (1), *A*_Control_ represents the absorbance of the control group at 600 nm, and *A*_SSPs_ represents the absorbance of the treated group at 600 nm. Furthermore, the antimicrobial activity of SSPs was detected by the agar diffusion method. A bacterial suspension of 100 µL was spread on an MHB agar medium (with 1.5% agar added to the MHB), and a sterile Oxford cup was placed on the surface and gently pressed. Then, 200 µL of SSPs liquids (filtered with a 0.22 μm filter membrane) was added to the Oxford cups, incubated at 37 °C for 24 h and then photographed to observe the inhibition zone on the surface of the medium.

### 3.4. Peptide Sequence Identification

The sequence of peptides was identified by LC-MS/MS technique [18]. Prior to the LC-MS/MS analysis, the samples were reduced and alkylated with dithiothreitol and iodoacetamide, then dissolved in a solution of (0.1% (*v*/*v*) formic acid (FA) and 5% (*v*/*v*) acetonitrile (ANC)). The samples were separated by chromatography using a chromatographic system with a nanoliter flow rate (Thermo Easy nLC, Thermo Scientific™ Technologies, Waltham, Massachusetts, USA). The chromatographic conditions were as follows: the flow rate was 500 nL/min over 40 min. The mobile phase A gradient was 0–3 min, 95–90%; 3–4.5 min, 90–72%; 4.5–28 min, 72–60%; 28–40 min, 60–0%. The mobile phase B gradient (0.1% FA, 80% ANC) was 0–3 min, 3%; 3–4.5 min, 3–8%; 4.5–28 min, 8–32%; 28–31 min, 32–44%; 31–34 min, 44–99%; 34–40 min, 99%.

After peptide separation, mass spectrometry was performed using a mass spectrometer (Q Exactive™, Thermo Scientific™ Technologies, Waltham, Massachusetts, USA). The spray voltage and capillary temperature were set to 2.2 kV and 270 °C. The detection mode was set to positive ion; the precursor ion scan range was 350–1550 *m*/*z*, the first-order mass spectrometry resolution was 120,000, the AGC target was 4e5 and the first-order mass spectrometry Maximum IT was 50 ms. For the peptide MS/MS analysis, the 20 highest-intensity precursor ion MS/MS spectra were triggered for acquisition after each full scan. The MS/MS resolution was 50,000, the AGC target was 1e5, the maximum IT was 100 ms, and the normalized collision energy was 32%.

A PEAKS Studio 10.0 search engine was used for manual de novo sequencing with the following parameters: Precursor Mass Tolerance: 15.0 ppm, Fragment Mass Tolerance: 0.03 Da, precursor mass search type: monoisotopic, max missed cleavages: 100 and max variable PTM per peptide: 3. The peptides obtained from the de novo sequencing analysis were identified and matched against the Uniprot database (https://www.uniprot.org/, accessed on 15 March 2023). Additionally, all identified peptides were compared using the CAMP_R4_ (http://www.camp.bicnirrh.res.in/, accessed on 15 March 2023) and results with higher scores and lower E-values were selected.

### 3.5. Virtual Screening

To screen potential AMPs, computational prediction tools were used [47]. Peptides with an amino acid number of less than or equal to 5 were eliminated, and the remaining peptides were screened (including two parts: the peptide sequences obtained from de novo sequencing and the peptide sequences identified based on the Uniprot database). First, CPPpred (http://distilldeep.ucd.ie/CPPpred/, accessed on 21 March 2023) was used to predict the cell-penetrating potential of the peptides, and the theoretical cell-penetrating activity was represented as a calculated score. The higher the score, the greater the potential. To reduce the probability of false positives, the CPPpred score threshold was set to be greater than 0.7 [48]. For the peptide sequences obtained from de novo sequencing, the CPPpred score of all peptides is less than 0.7, so this step of the CPPpred screening was skipped. Then, the peptides filtered in the previous step were further screened using the CAMPR4 (http://www.camp.bicnirrh.res.in/predict/ accessed on 21 March 2023) prediction tool (support vector machine and random forest algorithm). The filtering conditions were set to a support vector machine score greater than 0.5 and a random forest score greater than 0.45. The physicochemical properties of the candidate peptides were calculated, and their toxicity was predicted using the ToxinPred (https://webs.iiitd.edu.in/raghava/toxinpred/index.html, accessed on 21 March 2023).

### 3.6. Molecular Docking

Molecular docking was applied to estimate the interactions between peptide and receptors protein to screen for more potent AMPs [17]. The peptides’ two-dimensional (2D) structure was predicted using the ChemDraw software (Version 20.0, Wellesley, Massachusetts, USA), and then, its three-dimensional (3D) structures were generated through energy minimization. The X-ray structures of DNA gyrase and DHFR were obtained from the Protein Data Bank (https://www.rcsb.org/, accessed on 28 March 2023) and their PDB codes are 6RKS and 7REG, respectively. Candidate peptides were selected as ligands, and two key enzymes were selected as receptors. The positions of the ligands in the X-ray structures of DHFR and DNA gyrase were used as binding sites for molecular docking analysis. The small molecule ligands bound to the two key enzymes were removed along with hydrogen atoms using PyMol (Version 2.5.0, Schrödinger Inc., New York, NY, USA). AutoDockVina (Version 1.2.0, The Scripps Research Institute, La Jolla, CA, USA) was used for multiple molecular docking analyses of the peptides. The docking process used a flexible docking mode, where the side chains of the binding pocket amino acids could be optimally adjusted according to the ligand conformation. Based on the docking results, peptides with binding energies to DNA gyrase and DHFR of less than -5.5 kcal/mol were selected for the next analysis step. The interactions between the ligand peptide and the two receptor enzyme molecules were analyzed by LigPlot^+^ (Version 2.2, European Bioinformatics Institute, Cambridge, UK), while the best binding mode was selected using PyMol for visualization.

### 3.7. Peptide Synthesis

ChinaPeptides (QYAOBIO) Co., Ltd. (Shanghai, China) was commissioned to artificially synthesize the peptides obtained from the aforementioned molecular docking screening by solid-phase synthesis method [18,47]. The peptides were purified by High Performance Liquid Chromatography and characterized by electrospray ionization mass spectrometry to ensure their purity was above 95%.

### 3.8. Antibacterial Activity of Synthetic Peptides

The antimicrobial activity of the synthetic peptides was determined using a conventional broth microdilution method [18]. *E. coli* was cultured with MHB medium at 37 °C and 60 r/min for 16–24 h until it reached a viable bacteria count of approximately 10^7^ CFU/mL. The synthetic peptides were dissolved in double-distilled water and diluted in a 2-fold gradient using MHB medium in a 96-well plate, with final concentrations ranging from 10,000, 5,000, 2,500, 1250, 625, 312.5, 156.25… to 0 µg/mL, respectively. The control group did not contain synthetic peptides. Subsequently, 1 μL of *E. coli* suspension was added to each well and incubated at 37 °C for 24 h. The absorbance was measured at 600 nm using a microplate reader (Spark10M, Tecan, Mannedorf, Switzerland). The inhibition rate of the synthetic peptide against *E. coli* was calculated according to equation (1), and the half-maximal inhibitory concentration (IC_50_) was calculated by non-linear fitting.

### 3.9. Membrane Potential Assay

The membrane potential was determined according as described in our previous study [26]. As described in Section 3.8, *E. coli* was cultured to reach a viable bacteria count of approximately 10^7^ CFU/mL. The synthetic peptides were dissolved in double-distilled water and diluted to different concentrations using MHB medium. *E. coli* was treated with different concentrations of synthetic peptides and incubated at 60 r/min and 37 °C for 12 h. The same volume of MHB medium was used instead of synthetic peptide for the control group. After incubation, the bacterial suspension was centrifuged at 2000× *g* for 5 min and washed twice with 0.1 M PBS (pH 7.4). Subsequently, 20 µL of the fluorescent probe DiBAC4(3) and 180 µL of bacterial suspension were added to a black opaque 96-well plate (Nunc, Copenhagen, Denmark). After incubation for 15 min at 37 °C in the dark, the fluorescence intensity was measured at 490 nm excitation/525 nm emission wavelengths using a microplate reader (Spark10M, Tecan, Mannedorf, Switzerland).

### 3.10. Cell Viability Assay

Cell viability assay was performed according to Abdelrahman M. Qutb et al. [47]. As described in Section 3.8, *E. coli* was cultured to reach a viable bacteria count of approximately 107 CFU/mL. The bacterial suspension was centrifuged at 2000× *g* for 5 min, then it was washed twice with 0.1 M PBS (pH 7.4) and resuspended. 100 µL of the resuspension and 10 µL of PI fluorescent dye (50 μg/mL) were added to a black opaque 96-well plate (Nunc, Copenhagen, Denmark) and incubated at 37 °C for 6 h in the dark. Then, a drop of the suspension was transferred to a glass slide and observed using an inverted fluorescent microscope (Olympus IX73, Tokyo, Japan). The fluorescence intensity was measured at 535 nm excitation/617 nm emission wavelengths using a microplate reader (Spark10M, Tecan, Mannedorf, Switzerland).

### 3.11. Intracellular Localization of Peptides

The synthetic peptides were labeled with fluorescein isothiocyanate (FITC) by ChinaPeptides (QYAOBIO) Co., Ltd. (Shanghai, China). Intracellular localization of peptides was performed according to previous studies [47,49]. As described in Section 3.8, *E. coli* was cultured to reach a viable bacteria count of approximately 10^7^ CFU/mL. The FITC-labeled peptides were dissolved in double-distilled water or dimethyl sulfoxide (final concentration 0.25%) at a final concentration of 250 μg/mL. Two peptides were mixed in equal proportions at equal concentrations to obtain a final concentration of 125 µg/mL. *E. coli* was treated with the FITC-labeled peptides and incubated at 60 r/min and 37 °C for 3 h. The bacteria were collected by centrifugation and washed twice with 0.1 M PBS (pH 7.4) and the supernatant was discarded. The resulting bacteria were fixed with 4% paraformaldehyde for 30 min and centrifuged at 2000× *g* for 3–5 min. Then, the supernatant was discarded and the resulting bacteria were mixed with 400 μL DAPI and 100 μL PBS, left for 10–15 min in the dark, washed twice with PBS and resuspended. One drop of the bacterial solution was added to a microscope slide. After the mounting medium (G1401, Servicebio, China) treatment, LSCM (Nikon Eclipse Ti, Tokyo, Japan) was used to analyze the localization of the peptides in *E. coli* cells.

### 3.12. Statistical Analysis

All experiments were repeated three times to ensure the reliability of the data. The experimental data were expressed as mean ± standard deviations. The data were processed and analyzed using the SPSS software (Version 24.0, Chicago, IL, USA), and statistical analysis was carried out by one-way analysis of variance (ANOVA). Duncan’s test was used to determine the significance of the differences among the means (differences were considered statistically significant at *p* < 0.05). The IC_50_ was calculated based on the logistic function using the Origin software (Version 2019b, Northampton, MA, USA).

## 4. Conclusions

In this study, SSPs were found to exhibit inhibitory activity against *E. coli*, *S. aureus* and *P. aeruginosa*. The proteins matched with the SSPs were identified as histones of *A. ruthenus*, containing fragments of histone H1, H2A, H2B and H4. Histone H2B (45%) showed the highest coverage, followed by H2A (35%). Two novel peptides, NDEELNKLM and RSSKRRQ, which have excellent potential antibacterial activities, were screened by in silico prediction tools and molecular docking. They interact with DHFR and DNA gyrase by forming salt bridges and hydrogen bonds. Furthermore, NDEELNKLM and RSSKRRQ exhibited a superior antibacterial effect against *E. coli* when mixed in equal proportions at the same concentration and led to changes in cell membrane permeability and reduced cell viability. NDEELNKLM and RSSKRRQ could enter the interior of *E. coli* cells to exert antibacterial effects by affecting nucleic acid metabolism. These results provide theoretical support for the mining and development of novel AMPs.

## Figures and Tables

**Figure 1 marinedrugs-21-00386-f001:**
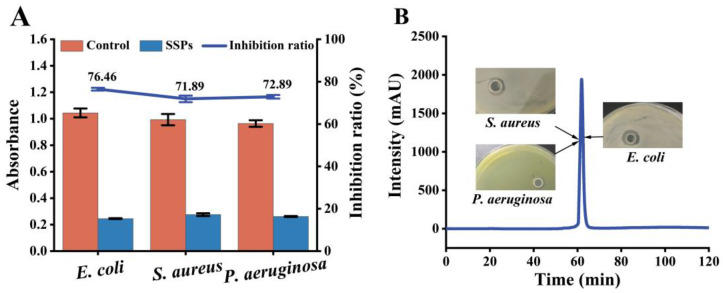
Antibacterial activity of SSPs before purification (**A**) and after purification (**B**). In Figure A, the histogram shows the absorbance of the bacterial suspensions, and the control group used the same volume of MHB medium instead of SSPs. In Figure B, the blue line shows the elution profiles of the SSPs during purification and the Oxford tower test was used to verify the antibacterial activity of the purified component.

**Figure 2 marinedrugs-21-00386-f002:**
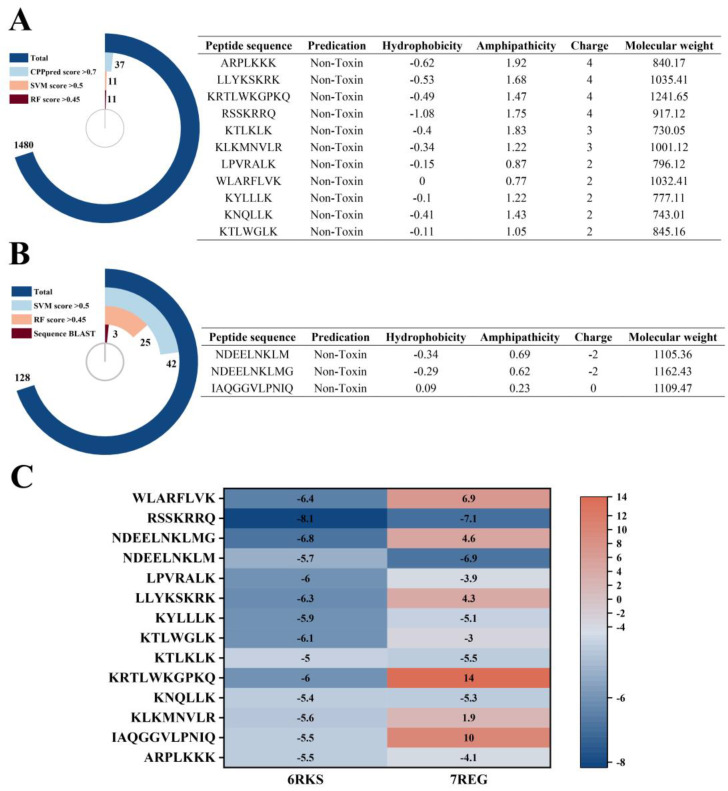
Virtual screening of AMPs. (**A**,**B**) represent the virtual screening results of AMPs based on de novo sequencing analysis and Uniprot database identification, respectively. The arc diagram shows the virtual screening process. According to the screening conditions, the simulation runs step by step from the outside of the arc to the inside, and the value indicates the number of peptides filtered in each step. The right side of the arc graph shows the physicochemical properties and toxicity prediction of the final screened candidate peptides. Sequence BLAST indicates the number of peptides after comparison with the CAMP_R4_ database. (**C**) shows the molecular docking score (binding energy (kcal/mol)) heat map of the screened candidate peptides with two receptor proteins (6RKS and 7REG).

**Figure 3 marinedrugs-21-00386-f003:**
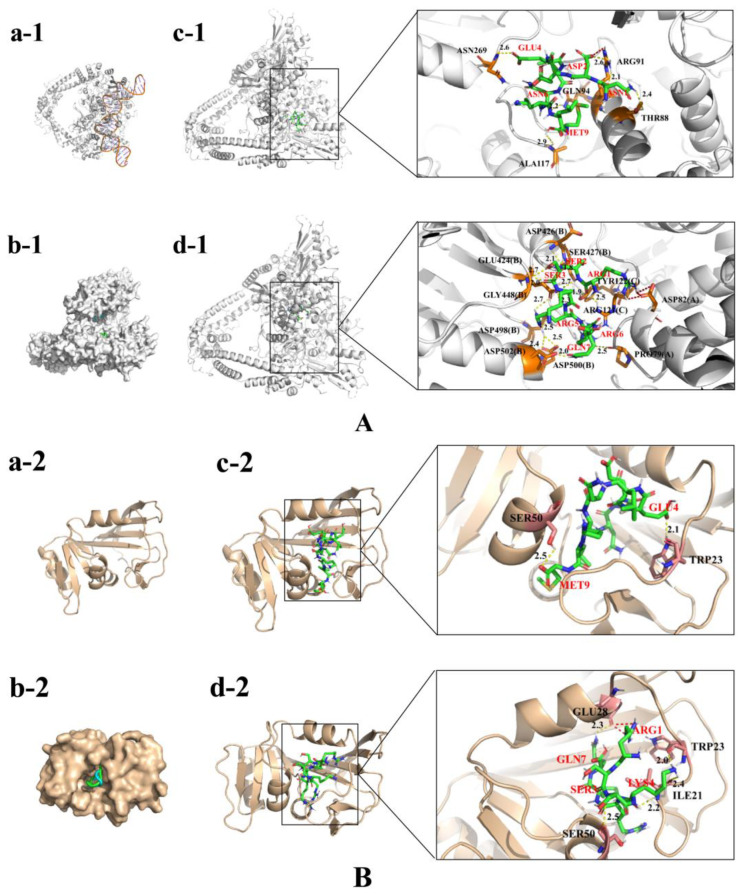
Molecular docking of peptides with DNA gyrase (**A**) and DHFR (**B**). The molecular surface of DNA gyrase is silvery-white, and that of DHFR is brown-yellow. **a-1** and **a-2** represent the Cartoon mode of DNA gyrase (in the binding state to both DNA strands) and DHFR. **b-1** shows the binding modes of NDEELNKLM and RSSKRRQ on the molecular surface of DNA gyrase. **b-2** shows the binding modes of NDEELNKLM and RSSKRRQ on the molecular surface of DHFR. DNA gyrase and DHFR are shown in surface mode, and the peptide molecules are shown in cartoon mode. **c-1** and **d-1** represent the 3D binding modes of NDEELNKLM and DNA gyrase, respectively. **c-2** and **d-2** represent the 3D binding modes of RSSKRRQ and DHFR, respectively. DNA gyrase and DHFR are shown in cartoon mode, and their residues are shown as brown-yellow and dark-red sticks. The amino acid names and sequence numbers indicating where the residues are located are shown in black, and the peptide molecules are shown as blue-green sticks. The yellow dashed lines represent hydrogen bonds, and the red dashed lines represent salt bridges. The marked numbers represent the bond lengths of the hydrogen bonds. The peptide molecules’ amino acid names and sequence numbers are shown in red.

**Figure 4 marinedrugs-21-00386-f004:**
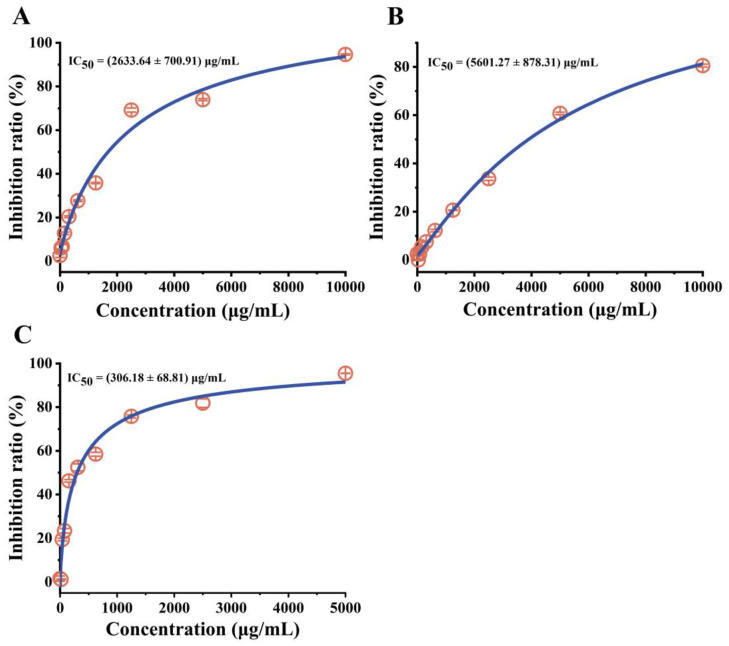
Antibacterial activity of the synthetic peptides against *E*. *coli*. (**A**): NDEELNKLM, (**B**): RSSKRRQ and (**C**): mixture of RSSKRRQ and NDEELNKLM.

**Figure 5 marinedrugs-21-00386-f005:**
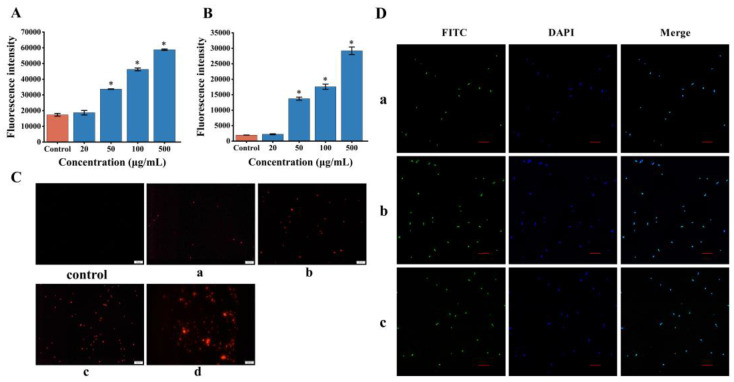
Analysis of the antibacterial mechanism of NDEELNKLM and RSSKRRQ against E. coli. (**A**): analysis of membrane potential of *E. coli*; (**B**,**C**): analysis of cell viability of *E. coli*; (**D**): analysis of peptides localization. In (**A**,**B**), the concentration represents the final concentration of the mixed peptides, and * indicates a significant difference in fluorescence intensity compared to the control group (*p* < 0.05). In Figure (**C**), **a**–**d** represent the concentrations of the mixed peptides of 20, 50, 100 and 500 μg/mL, respectively. The control groups in Figures (**A**–**C**) do not contain peptides. Figure (**D**) shows the LSCM images of E. coli treated with different peptides (green: FITC, blue: DAPI), **a**: NDEELNKLM, **b**: RSSKRRQ and **c**: a mixture of NDEELNKLM and RSSKRRQ. The image scale is 10 μm.

**Table 1 marinedrugs-21-00386-t001:** The matched histone information based on the Uniprot database.

No.	Accession	Score	Coverage (%)	Peptides	Average Mass (Da)	Description
1	A0A444UNK7	137.49	35	25	27,177	*A. ruthenus* Histone H2A
2	A0A444UNJ1	122.75	30	14	13,729	*A. ruthenus* Histone H2A
3	A0A444UBZ7	93.95	11	4	13,599	*A. ruthenus* Histone H2A
4	A0A444UFZ9	80.73	45	10	13,758	*A. ruthenus* Histone H2B
5	A0A444UNI9	80.73	44	10	13,847	*A. ruthenus* Histone H2B
6	A0A662YT88	76.31	17	7	20,965	*A. ruthenus* Histone H1
7	A0A444UNK9	51.66	28	7	11,349	*A. ruthenus* Histone H4
8	A0A444UNI6	51.66	28	7	11,367	*A. ruthenus* Histone H4
9	A0A444UKX4	51.66	27	6	11,448	*A. ruthenus* Histone H4
10	A0A444V4L3	51.66	20	5	11,411	*A. ruthenus* Histone H4
11	A0A444UV61	51.66	17	3	11,379	*A. ruthenus* Histone H4

**Table 2 marinedrugs-21-00386-t002:** Blast results of the identified peptides based on the CAMP_R4_.

No.	Peptide	Blast Results of CAMP_R4_				
		Organism	Description	Sequence Length	Antibacterial	Reference
1	KESYAIY	*Rhacophorus schlegelii*	H2B	125	Gram-negative (*E. coli*)	[31]
2	NSFVNDIFE					
3	YNKRSTITS					
4	NDEELNK	*Litopenaeus vannamei*	H2A	122	Gram-positive (*Micrococcus luteus*)	[32]
5	NDEELNKL					
6	NDEELNKLM					
7	NDEELNKLMG					
8	IAQGGVLPNIQ					
9	GGVLPNIQA					
10	GGVLPNIQ					
11	GVLPNIQ					

## Data Availability

Data are contained within the article.

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
