# Peer review of "Identification, Screening and Antibacterial Mechanism Analysis of Novel Antimicrobial Peptides from Sturgeon (Acipenser ruthenus) Spermary"

_marinedrugs, 2023, doi:10.3390/md21070386_

Round 1

Reviewer 1 Report

The article describes the enzymatic production of peptides derived from sturgeon spermary extracts, which in a previous work (Front. Nutr. 2022, 9, 1021338) had shown antimicrobial activity due to their histone content. The antimicrobial activity of the peptides was assayed, and their correlation with the sequence of known histones was explored. The mode of action of the most active peptides was also studied, and their potential interaction with biological targets such as E.coli DNA gyrase was studied with molecular modelling.

The article is well written and contains much work, but in spite of this, the results are not particularly interesting, since the most active peptides have poor antimicrobial activity (IC50 = 2633,64 µg/mL and IC50 = 560103,64 µg/mL). Although their combination reduces this ratio to IC50 = 306 µg/mL, this is still a high value, compared with those of other extracts and compounds. The use of enzymatic hydrolysis would not be necessary since the previously reported crude spermary extract displayed significant antimicrobial activity.

Therefore, the article is recommended for other Journal, more focused on peptides or antimicrobial peptides, where the structure-activity relationships would be discussed.

In addition, several things should be taken into account:

1)      The references on spermary antimicrobial peptides (page 2, line 60) should be grouped and added. In addition to current references 17 (previous work), 20 etc, recent related work on antimicrobial histones from salmon testis should be included: Mar. Drugs, 2020, 18, 133.

2)      Are the IC50 in Figure 4 relative values (otherwise the IC50 values would not match the graphics)? Which values were provided to the Origen software?

3)      The colocalization studies in Fig 5 should be supported by a higher-quality photo, the current one has low resolution and only a black image can be seen in the squares. The green and blue dyes are not seen against the black background.

Author Response

Thank you for your comments. We have made modifications in the revised manuscript, please see the attachment.

Reviewer 2 Report

1- Rewrite the abstract section to include the introduction, methods, results, and conclusion.

2- E. coli and any other name should be written as full names when written to first once.

3- In line 43, in vivo, in vitro should be written italic.

4- In results section, You recorded that the inhibition rate against E. coli reached 76%, but the inhibition zone does not represent this result; inhibition zones are very little less than 8 mm, so I believe there is a mistake. in case of pseudomonas have not inhibition zone, ???????

5- In line 82, you should not use strong or large, but rather values of the results, and the inhibition zones should also be represented as values. 

6- in line 74, you stated that The results demonstrate that the SSPs significantly inhibit the growth of E. coli , you should mention degree of significance P = .........?

7- Figure 1 caption should be clarified. Figure 1B also fails to describe what advantages it provides. How do you compare before and after purification, as the methods used differ?

8-All figure captions should be rewritten to be more concise and understandable.

9- There are two types of histone H2B in table 1, and you recorded three from where in line 106. If you have previous research, you should mention Rhacophorus schlegelii rather than A. ruthenus.

10- in line 264, rephrase Fig. 5 to Figure 5

11. in line 284, and in whole manuscript uniform SSPs or SSPE

12. in line 288, how enzymatic reaction is at boiling point, the enzyme denaturation occurred?

13. in line 300, did use two cultures or three culture, staphylo. E. coli. Pseudomanas

14. in line 302, treated by what

15. in line 304, which the sample? are these samples before purification and separation of peptides?

16. line 305, change cultured to incubated

17. In material and methods, 3.4. Peptide sequence identification, add a references.

18. In material and methods, 3.5. Virtual screening, add a references.

19. In material and methods, 3.6. Molecular docking, add a references.

20. In material and methods, 3.7. Peptide synthesis, add a references.

21. In material and methods, 3.8. Antibacterial activity of synthetic peptides, 3.9. Membrane potential assay, 3.10. Cell viability assay and 3.11. Intracellular localization of peptides, add a references.

 Minor editing of English language required

Author Response

First of all, thank you for your recognition of the research content of the paper, and then according to the valuable comments, we have made modifications in the revised manuscript, please see the attachment.

Reviewer 3 Report

In this study, the researchers aimed to discover and evaluate antibacterial peptides derived from sturgeon spermary peptides (SSPs) for their antibacterial activity and mechanism. The results showed that SSPs exhibited strong inhibitory activity against Escherichia coli, with an inhibitory rate of 76.46%. The majority of SSPs were derived from histones of the sturgeon species, with histone H2B having the highest representation (45%). Through in silico prediction tools and molecular docking, two novel peptides were found to interact with E. coli's DNA gyrase and dihydrofolate reductase. When the peptides were combined, their antibacterial effect was enhanced compared to individual peptides. The two novel peptides were also observed to alter the permeability of E. coli cell membranes, potentially exerting antimicrobial activity by inhibiting nucleic acid metabolic processes.

Major comments:

1.     In this study the authors present histone-related fragments (papain cleaved) that carry AMP function. However, it is quite difficult to fully understand what is the actual composition of the sturgeon spermary peptide extract. The complexity of the extract was not presented (only summary statistics is available, but no data is accessible). For example, the MS results of peptides that are not found in CAMPR4. E.g., accessibility of data reported in Table 1. (e.g. A0A444UNK7). Provide missing data.

2.     Search BLAST on UniprotKB and not only on CAMPR4. It may provide information and quality control for identified peptides that match other organisms.

3.     The rational for testing two receptors of DNA gyrase and DHFR of E. coli needs stronger justification. What evidence claim that they are the genuine targets for AMPs? Using docking of these two receptors but use the proteins of the other sensitive bacteria (as in Figure 1) can add to the confidence that these receptors are indeed likely targets.

4.     Testing peptides that fail to act as AMPs (I.e. by creating a synthetic mutated peptide) is needed to define specificity. Will be useful to include a negative result for bacterium that is not inhibited by the extract (if there is one like that). On the same spirit, can the extract lose the inhibitory activity (by changing PH / modifying the peptide). without showing such sensitivity for the peptides, specificity is impossible to assess.

Minor comments

1.      In introduction. Explain whether the histone-like AMP are fish/toad specific. accurate background information (e.g., Tanaka et al Antibiotics 2022, 11(9), 1240)

2.     The initial discussion concerns the extracellular histone and act on membrane/ membrane potential. The other part deals with intracellular ‘binding’ to receptor (a low affinity). Are all identified peptides act with both mode of action. Please clarify.

3.     Line 42: AMPs are not structurally conserved class. AMPs are very diverse. Please revise.  Line 47: Be precise, Histones are not defined by being part of the innate immune.

4.     Explain the score of Table 1. What score values are considered reliable?

5. When species are mentioned better to add 'common name' (like frog, shrimp)

Writing is smooth and relatively easy to read. 

Author Response

(The authors gave the same response as above.)

Reviewer 4 Report

Identification, screening antibacterial mechanism analysis of novel antimicrobial peptides from sturgeon (Acipenser ruthenus) spermary. 

The manuscript reflects a lot of work to identify antimicrobial peptides present in spermary sturgeon. The methodology applied is well explained.
It is intriguing because in the docking analysis it checks for only two enzymes without prior evidence that the peptides act on those enzymes and subsequently appears to be correct.
The understanding of the article would be aided by making the first paragraphs of the Introduction more appropriate and by adding a separate Discussion section.

The data in Figures 1A and 1B cannot be compared when using different techniques. Furthermore, the footnote of Fig. 1 does not specify which peptides are purified. It should indicate that in 1B Oxford towers are being used.
The genus name is missing in P. olivaceus and L. vannamei.
In Figure 5C and 5D the photos are not clearly seen.

Author Response

(The authors gave the same response as above.)

Round 2

Reviewer 1 Report

The authors have carried out the suggested modifications, and therefore the manuscript meets the criteria to be published in Marine Drugs. Only two minor types in the abstract: in line 6 remove one of the "by", in line 4 should say "were used to analyze and identify". The squares in Figure 5 are still somewhat dark, but the quality has improved (and probably the production department can increase the contrast/brightness further).

Author Response

Point 1: in line 6 remove one of the "by", in line 4 should say "were used to analyze and identify".

Response 1: Thank you for your careful check. The mistakes have been corrected in revised manuscript. See page 1 line 20, 22.

Point 2:The squares in Figure 5 are still somewhat dark, but the quality has improved (and probably the production department can increase the contrast/brightness further).

Response 2: Thank you for your suggestion. The contrast/brightness of Figure 5 was increased by Adobe Illustrator and Photoshop tools, and the resolution was increased to further improve the image quality and make the image clearer.

Reviewer 2 Report

Accept in present form

Minor editing of English language required

Author Response

Point 1: Minor editing of English language required

Response 1: We are grateful to reviewer 2 for his/her effort reviewing our paper and positive feedback. We carefully checked the English language of the manuscript and made revisions. See page 1 line 20, 22, 26; page 8 line 252; page 9 line 259, 261; page 12 line 420.

Reviewer 3 Report

In this study, the researchers aimed to discover and evaluate antibacterial peptides derived from sturgeon spermary peptides (SSPs) for their antibacterial activity and mechanism. The authors tried to reply to most comments, some of the comments were not fully addressed but will be considered in future works. It will be good to provide in supplemental Tables missing data from the MS.

I find that the revised version to be more complete. It shares rich information that could make it an interesting addition to  AMP collection.  

Minor comments:

1.     Line 66: Replace Virtual by virtual.

2.     Line 144: I still find it confusing to announce histones as AMP.

It is best to say that the histones have a crucial role in the packaging and organization of DNA and chromatin within the nucleus of cells. These are conserved, positively charged proteins. In several systems, it was also shown that histone-derived fragments carry antimicrobial activities in a diverse range of organisms (frog, shrimp etc).

3.     Line 264: Please explain “Oxford towers were used to verify the antibacterial ..”

4.     Line 867: Replace ‘better’ with more informative information (more potent? more specific?’ more stable?).

5.     Improve English in line 958

6.     Figure 1 A is Figure 1A, same for Figure 1B

I believe it is publishable (after some rephrasing and English improvement) 

Author Response

Thank you for the positive comments on the revised version. The issues raised have been solved one by one as detailed below.

1.  Line 66: Replace Virtual by virtual.

Response: Thank you for your careful check. It has been revised. The line number you marked is not quite correct. I think it should be page 1 line 21.

2.  Line 144: I still find it confusing to announce histones as AMP.

It is best to say that the histones have a crucial role in the packaging and organization of DNA and chromatin within the nucleus of cells. These are conserved, positively charged proteins. In several systems, it was also shown that histone-derived fragments carry antimicrobial activities in a diverse range of organisms (frog, shrimp etc).

Response: It is very grateful for your suggestion. The unclear description has been revised and improved in revised manuscript. See page 2 line 51-55.

3.  Line 264: Please explain “Oxford towers were used to verify the antibacterial ..”

Response: Thank you for your careful check. The meaning of this sentence was to verify the antibacterial activity of purified components by the Oxford tower test. The inappropriate sentence has been corrected in revised manuscript. See page 3 line 100-101.

4.  Line 867: Replace ‘better’ with more informative information (more potent? more specific?’ more stable?).

Response: Thank you for your suggestion. I think it should be line 373. It has been replaced by more potent. See page 11 line 375 in revised manuscript.

5.  Improve English in line 958

Response: Thank you. But the line number you marked is not quite correct. It's hard to find the sentence.

6.  Figure 1 A is Figure 1A, same for Figure 1B

Response: Thank you for your suggestion. The mistake has been corrected in revised manuscript. See page 2 line 88.